# Chemical Modification of Aptamers for Increased Binding Affinity in Diagnostic Applications: Current Status and Future Prospects

**DOI:** 10.3390/ijms21124522

**Published:** 2020-06-25

**Authors:** Jan P. Elskens, Joke M. Elskens, Annemieke Madder

**Affiliations:** Department of Organic and Macromolecular Chemistry, Faculty of Sciences, Ghent University, Krijgslaan 281 S4, 9000 Ghent, Belgium; Jan.Elskens@UGent.be (J.P.E.); Joke.Elskens@UGent.be (J.M.E.)

**Keywords:** aptamer, diagnostic applications, chemical modifications, enhanced binding affinity, SELEX

## Abstract

Aptamers are short single stranded DNA or RNA oligonucleotides that can recognize analytes with extraordinary target selectivity and affinity. Despite their promising properties and diagnostic potential, the number of commercial applications remains scarce. In order to endow them with novel recognition motifs and enhanced properties, chemical modification of aptamers has been pursued. This review focuses on chemical modifications, aimed at increasing the binding affinity for the aptamer’s target either in a non-covalent or covalent fashion, hereby improving their application potential in a diagnostic context. An overview of current methodologies will be given, thereby distinguishing between pre- and post-SELEX (Systematic Evolution of Ligands by Exponential Enrichment) modifications.

## 1. Introduction

Aptamers are single stranded DNA or RNA oligonucleotides, which act as synthetic receptors for the recognition of analytes and exhibit high target selectivity and affinity. These properties are driven by the secondary and tertiary structure of aptamers, including motifs such as loops, bulges, hairpins and pseudoknots [1]. This allows aptamers to exist in a wealth of shapes and forms, explaining their remarkable ability to recognize a wide variety of analytes. The molecular shape complementarity between the aptamer and its ligand, together with the presence of intermolecular aptamer–ligand contacts (hydrogen bonding, dipole–dipole interactions, ion–ion interactions, London dispersion forces and aromatic π–π stacking) are of crucial importance to obtain high affinity binding [2,3].

The outstanding molecular recognition between aptamers and their targets makes them powerful tools in diagnostic applications. As a consequence, the development of so-called aptasensors has received considerable attention, which has resulted in an extensive variety of detection strategies, ranging from electrochemical, colorimetric to gravimetric detection [4]. These aptamer-based biosensors use aptamers to recognize and detect various analytes such as cocaine, antibiotics and disease biomarkers, to name just a few [5,6,7]. Furthermore, aptamer-linked immobilized sorbent assays (ALISA) and aptamer-based lateral flow immunoassays have been developed [8,9]. The use of electrochemical sensing, for example, in the detection of the antibiotic ofloxacin, has also been reported [5]. Additionally, aptamers decorated with a suitable reporter group (fluorescent moiety, radioactive label, etc.,) have been used for the localization of target tissue and tumors, the monitoring of biological processes and the early diagnosis of diseases [10,11]. Traditionally, these applications relied on antibodies for the recognition of the analyte.

Due to their three-dimensional folding and target recognition, aptamers are often referred to as ‘chemical antibodies’ [12]. Hence, they can be regarded as promising substitutes for antibodies in diagnostic applications, especially in view of some of the disadvantages of the latter. Indeed, the most prominent method for generating antibodies relies on the immunization of animals with an immunogenic analyte. Small analytes are typically non-immunogenic due to their low molecular weight, thus, rendering the generation of suitable antibodies difficult, and while methods have been developed to remediate this drawback, these procedures are costly and time consuming [13,14]. Aptamers, on the other hand, do not suffer from this limitation and can be generated against a wide range of analytes, such as ions [15], small molecules [16], proteins [17], viruses [18], bacteria [19] and even whole cells [20]. Furthermore, the higher physical and temperature stability of aptamers, as compared to antibodies, results in a more robust analytical tool. Even after denaturation, given the reversibility and predictability of base-pairing, aptamers are able to refold at lower temperatures, which restores their properties and increases their shelf life [21]. In contrast, antibodies are unstable under most conditions deviating from physiological ones and refolding of antibodies after denaturation may be difficult to achieve. From a production point of view, antibodies are prepared using cell lines, which makes them highly priced products. Moreover, batch-to-batch variation is observed for polyclonal antibodies. This is in strong contrast to the production of aptamers, which is rather straightforward and delivers constant quality as it relies on reproducible solid phase or enzymatic synthesis. This results in a more cost-efficient production process, leading to cheaper diagnostic tests and cost-effective healthcare. 

Finally, a major advantage of aptamers over antibodies is the possibility for selective chemical modification. The labeling of antibodies or aptamers is frequently required for certain applications (e.g., antibody–drug conjugates, labeled antibodies for immunosensors, etc.). However, the site-selective labeling of antibodies is particularly difficult and is restricted to a small set of possible chemical reactions. Moreover, poorly controlled labeling can lead to conformational changes and loss of binding affinity [22]. Because aptamers are synthesized chemically, they can easily be modified selectively with an almost endless variation in accessible structures. 

Aptamer modification can allow an increase in binding affinity with its target in two distinct ways. As nucleic acids are composed of a concatenation of four different nucleotides (A, C, G, T/U), the chemical variety that is naturally present is far more restricted compared to proteins with an option of 20 different amino acids. This could limit the full application potential of aptamers as the binding interactions are not always optimized for their ligand. By chemically modifying the aptamer, the functional groups that can be introduced to interact with the ligand, can be diverse and even easily surpass the diversity of the 20 naturally occurring amino acids. A second way to increase the binding affinity takes the dynamic nature of aptamers into account (Figure 1). Generally, an equilibrium exists between the unfolded state (I) and the binding competent (folded) state (II), with a corresponding equilibrium constant K_F_. The binding competent state (II) is responsible for binding the target with a dissociation constant K_D_. The actual conformation, and thus, binding ability of aptamers can be influenced by several factors, such as salt concentration, pH, temperature. Inevitably, not all aptamer conformations are capable of binding the target. By stabilizing the aptamers in their binding competent state (II) via the introduction of suitable chemical modifications, the affinity of the aptamers for their ligands can be increased. 

In this review, an overview of the available methods for improving the binding affinity of aptamers with emphasis on diagnostic applications is provided (as opposed to their therapeutic potential [23,24,25,26]). First, we briefly describe the conventional SELEX procedure for aptamer selection together with pre- and post-SELEX modification strategies. In the subsequent section, we discuss aptamer modifications implemented in a diagnostic context. For this purpose, we have classified them into: (1) modified aptamers for enhanced non-covalent target interaction and (2) modified aptamers for covalent target trapping. Additionally, we take a glimpse into the future and elaborate on modifications that have proven successful in enhancing the binding affinity of aptamers but have not yet been used in a diagnostic context.

## 2. Methods for Introducing Chemical Modifications into Aptamers

### 2.1. The SELEX Strategy for Aptamer Discovery

Discovery of new aptamer sequences with affinity for a target of interest is typically achieved by screening a random library of chemically synthesized oligonucleotides (10^13^–10^15^ sequences) via a process called Systematic Evolution of Ligands by Exponential Enrichment (SELEX). This method was first published in 1990 by two independent research groups for screening a library of RNA sequences [27,28] and has since rapidly evolved as the method of choice for both DNA and RNA aptamer selection. The conventional SELEX procedure (Figure 2) starts with the synthesis of a randomized library of DNA sequences, which are flanked by primer regions necessary for further amplification via the polymerase chain reaction (PCR). For the screening of RNA aptamers, the DNA library should first be converted into RNA sequences via in vitro transcription. In the next step, the target molecule is incubated with the synthetic library (step 1), after which, a first selection round can be performed (step 2). A key aspect, herein, is the efficient separation of target-bound sequences from unbound sequences followed by amplification of the former via PCR (DNA aptamers) or reverse transcription-PCR (RNA aptamers) (step 3). The enriched library pool is then used for a next screening round (step 4). This process is repeated until the initial library is reduced to a considerable number of high affinity aptamers, obtained by steadily increasing the stringency of the selection process in each cycle. After the last cycle, the enriched aptamer pool is sequenced, and additional assays can be performed to characterize the binding behavior of the aptamer to its target. In recent years, multiple adaptations to the original protocol have been described for improving the aptamer selection process and to allow the screening of modified aptamers [29,30,31,32,33]. 

The limited functional diversity of aptamers compared to proteins may lead to suboptimal target binding. Therefore, it has been realized that extending or fine-tuning the range of possible intermolecular interactions between the aptamer and its target by chemical modifications can be an elegant and efficient way to enhance the stability and binding affinity of the aptamers. These extra chemical functionalities can be introduced in the aptamers before or after the SELEX process. In this review, limitations and advantages of pre- and post-SELEX derivatization will be discussed. A list of the aptamer modifications relevant to this review can be found in Table 1. 

### 2.2. Pre-SELEX vs. Post-SELEX Modifications

To date, multiple aptamer modifications have been introduced before or after the SELEX process in order to fine-tune aptamer properties (Table 1) [34]. As each strategy has its advantages, ideally, both are combined to obtain higher affinity aptamers compared to what can be achieved with one approach alone. For example, the affinity of C5-modified aptamers selected against the platelet-derived growth factor B could be further increased by post-SELEX modification [35]. For this purpose, the pre-SELEX incorporated C5-benzyl modifications were consistently substituted with alternative C5 moieties. Substitutions that resulted in increased binding were combined to further improve the affinity of the aptamer for the target. A similar approach was exemplified for post-SELEX improvement of a C5-modified aptamer against interleukin-6 [36]. Besides C5 substitutions also phosphodiester modifications, sugar modifications and spacers were introduced post-SELEX which led to further improvement of affinity, specificity and nuclease stability of the aptamer. It lies beyond the scope of this review to give a complete and extensive overview of all reported aptamer modifications. Instead, we will limit ourselves to the most representative examples and focus on their added value in diagnostic applications.

The most straightforward method for introducing modifications, either via solid phase synthesis or enzymatically, is by chemically modifying the existing nucleotide building blocks. A variety of functional groups, including hydrophobic, hydrophilic or charged moieties, can be incorporated via well-established chemistries and many of those building blocks are commercially available today. In this way, the aptamer modifications can be perfectly tailored, based on the structural requirements and features of the target. An important prerequisite when considering pre-SELEX aptamer modification, is that the envisaged modifications should be compatible with the polymerase enzymes used in the SELEX process. Extensive studies on polymerase activity when using non-natural nucleotides resulted in the identification of a series of structural requirements that modified nucleotides should obey in order to be compatible with the natural polymerase enzymes [37]. Firstly, the base pairing ability between modified nucleobases is not a mandatory prerequisite for substrate recognition by polymerases. More importantly is the ability for hydrophobic interactions between the modified nucleobase and the amino acids in the polymerase active site and the shape of the modified nucleobase, which should mimic the geometry of natural nucleobases for successful recognition and incorporation by the polymerase. Secondly, interactions between the amino acids in the polymerase active site and the nucleobases in the minor groove should preferentially be maintained as they often function as recognition elements for the polymerase. For example, the electron density imposed by the electron pair located on the N3 position in purines and the ketogroup at position 2 in pyrimidines was shown to be an important hydrogen bonding interaction site [34,38,39]. Modifications at the C5 position of pyrimidines and N7 position of purines are in general well tolerated as they do not interfere with the canonical hydrogen bonding and minor groove interactions. Besides structural optimization of the non-natural nucleotides, efforts have also been undertaken in the engineering of polymerases. Throughout the years, variants of different polymerases have been designed that allow incorporation of non-natural nucleotides with high tolerance in substrate structure [40,41]. An example is the mutant form of the terminator DNA polymerase that is able to incorporate C5-modified nucleotides at higher efficiency compared to the natural polymerase [42].

In the post-SELEX approach, modification is introduced after the SELEX process. In this case, the initial random DNA or RNA library is already narrowed down to a pool of high affinity aptamers before derivatization, which makes this strategy ideal for the synthesis of a broad variety of chemically modified aptamers. A major advantage of this approach is the fact that the envisaged extra chemical functionalities should not necessarily be compatible with the polymerase enzyme [43]. However, a common pitfall is that any modification introduced after the SELEX process may influence the binding affinity of the aptamer. For example, certain chemical modifications may alter the aptamer fold with concomitant negative consequences on target binding [24]. Therefore, typical post-SELEX modifications encompass the truncation of aptamers or the design of multivalent aptamers (vide infra). Post-SELEX modifications can also encompass conjugation reactions such as PEGylation or chemical stabilization of the aptamer against nucleases, for instance, achieved via the introduction of stable hairpin DNA in combination with reinforcement of the aptamer stem region [44]. Crosslink moieties are also often incorporated after the SELEX screening process. Whether introduced internally or at terminal positions of the aptamer sequence, these crosslink moieties impose minimal effects on the aptamer folding and binding event, which renders them suitable candidates for post-SELEX incorporation. Modifications with a higher chance of inducing structural changes such as modifications of the phosphodiester linkage, sugar or nucleobase modifications are often introduced in a pre-SELEX fashion, as in this case, the effect of post-SELEX modification on the aptamer function is much harder to predict. Nevertheless, when introduced on specifically chosen positions, post-SELEX incorporation of these modifications can enhance the aptamer binding properties considerably [35,36,45]. Figure 3 depicts a flow diagram of modifications elaborated upon in this review and their potential way of introduction before (pre) of after (post) the SELEX process.

## 3. Strategies for Improving Binding Affinity Through Chemical Aptamer Modification: Non-Covalent Target Binding

### 3.1. Modified Aptamers for Enhanced Non-Covalent Target Interaction

Enhancing binding affinity can be achieved by improving non-covalent interactions with the target analyte and the methodologies applied in this context can be subdivided into three main categories. The first class encompasses optimization (often downsizing by truncation) of the aptamer structure, which is a typical post-SELEX procedure. The second category is dedicated to the conjugation of multiple aptamers, which is an extremely powerful post-SELEX technique for increasing the binding strength towards a multimeric target. Finally, stronger binding aptamers can be generated by introducing specific chemical functionalities either in a pre- or post-SELEX set-up.

#### 3.1.1. Truncated Aptamers

Aptamers obtained after the SELEX process often require further optimization as they consist, in general, of approximately 80–100 nucleotides, which increases production cost and lowers the overall synthesis yield. The aptamer sequences normally encompass three categories of nucleotides: essential-, supporting- and nonessential nucleotides. The first two categories are crucial for target binding as they are responsible for target–aptamer interactions and the formation of secondary/tertiary structures respectively. Nonessential nucleotides, on the other hand, often belong to primer regions and tail sequences and are not involved in intramolecular interactions. Elington et al. previously stated that the constant primer regions have a minimal contribution to the structure and the binding event, which means they can often be removed without altering the binding properties of the aptamer [46]. Moreover, removing nonessential nucleotides (a process called downstream truncation) can have a beneficial effect on the binding characteristics, since these nucleotides can interfere with the target–aptamer interaction. In general, the truncation experiments start with the computational prediction of the folding and hybridization pattern of the selected aptamer. Afterwards, nonessential nucleotides are removed, thereby aiming to isolate the key binding motif, which is followed by evaluating the binding characteristics of the truncated aptamers. This simple but effective strategy has, for example, already proven to be widely applicable resulting in the successful truncation of a 56-mer VEGF-165 binding aptamer (a cancer biomarker) to a 23-mer sequence, which resulted in a 200-fold increase in binding affinity [47]. These truncated sequences act as the foundation upon which further modifications can be incorporated for the additional fine-tuning of aptamer properties [30,48,49]. Truncated aptamers have been successfully applied in diagnostic applications, which can be subdivided into three main categories ranging from (1) biosensors to (2) cell imaging and (3) enzyme-linked oligonucleotide assays (ELONA).

An example of the first category can be found in colorimetric assays, where aptamers are conjugated to gold nanoparticles. Binding of the target to the aptamer results in a conformational change, which induces nanoparticle aggregation and concomitant change in the color of the gold nanoparticles. This methodology has successfully been implemented for the sensitive detection of the pesticide acetamiprid, bisphenol A, the polychlorinated biphenyl PCB 77 and the antibiotic streptomycin [50,51,52,53]. Electrochemical biosensors have also been developed by conjugating aptamers on the electrode surface. After target binding, the structural conformation of the aptamer changes, which alters the impedance of the electrode and allows detection of the analyte. These aptamer-based biosensors have been used for the detection of glycated hemoglobin (an important diagnostic marker in diabetes) and the antibiotic tobramycin, amongst others [54,55]. Truncated aptamers have also been implemented in an optical biolayer interferometry (BLI)-based aptasensor for the detection of the marine neurotoxin Gonyautoxin 1/4 [49]. Upon binding of the neurotoxin to the surface immobilized aptamer, a change in the light interference pattern is observed, which can be used to detect the toxin in real-time. Another sensing system frequently employed in conjunction with truncated aptamers is fluorescent-based detection. In this case, a conformational change induced by binding of the target to the aptamer results in an alteration of the fluorescent signal. Both fluorescently-labeled and label-free biosensing approaches have been reported with the possibility of on-site detection using portable analyzers [56,57,58,59]. The target of interest can range from small molecules such as the mycotoxins aflatoxin B1 [60] and ochratoxin A [61], bisphenol A [62], the antibiotic tobramycin [54] and the marine toxin okadaic acid [63], to larger entities such as lipopolysaccharides [64] and the cyclic heptapeptide microcystin-LR (a hepatotoxin produced by cyanobacteria) [65]. Furthermore, proteins such as tropomyosin (a major shrimp allergen) [66] and glycated hemoglobin [67] or even whole bacteria such as *Salmonella enteritidis* [68] can be measured using this approach. A final example of a biosensor employing truncated aptamers is the surface-enhanced Raman spectroscopy-based aptasensors for the detection of the EpCaM protein (a cancer biomarker). This sensor allows the sensitive detection on a single cell level of early stages of cancer and has successfully been used for quantifying EpCaM, both in solution and in membrane-embedded cancer cells [69].

Next, truncated aptamers have successfully been implemented for the selective imaging of various cancer cells. For example, colon- and metastatic cancers were visualized using truncated aptamers labeled with the fluorescent dyes PE-Texas red and 6-fluorescein (FAM) respectively [47,70]. Moreover, using cells as the target in the SELEX process (so-called cell-SELEX) allowed the identification of truncated aptamers, which have been found to distinguish between poorly differentiated gastric cancer cells and healthy tissue. This is of great value, since these aptamers can be employed to detect gastric cancer in an early stage, thereby significantly improving the survival rate of the patient [71].

Finally, a last field of interest for the application of truncated aptamers is the development of enzyme-linked oligonucleotide assays (ELONA). This type of assay has proven pivotal for the early stage diagnosis of hepatocellular carcinoma, which is the most common type of primary liver cancer. Patients suffering from this disease are frequently diagnosed at a later stage, which drastically reduces their survival rate. Dickkopf-1 (DKK-1) is a serum protein biomarker for this disease and was considered a valuable target for the development of truncated aptamers. By implementing anti-DKK-1 aptamers in an ELONA set-up, a rapid test for the diagnosis of hepatocellular carcinoma could be obtained [72]. Clinically relevant aptamers have also been prepared against protein A (a cell-surface protein of the bacterium *Staphylococcus aureus*) and against the inactivated H1N1 virus [73,74]. Both aptamers have been successfully employed in ELONA assays for the efficient and fast detection of these pathogens.

#### 3.1.2. Joining Binding Motifs: Bivalent and Multivalent Aptamer Constructs

Most methods described in this review aim to increase the binding affinity of aptamers by optimizing the intermolecular interactions with their targets. However, an often forgotten and equally useful technique relies on the concept of avidity. When a molecule contains two identical binding sites, the binding affinity is increased compared to a molecule containing only one binding site, despite the fact that the binding affinity of all sites remains the same. The reason for this behavior is that a larger number of binding sites makes it more likely for the target to bind to the aptamer. Likewise, when an analyte contains multiple targets (for example, sugar moieties on the cell surface), a multimeric aptamer (possessing multiple copies of sugar-recognizing binding motifs) will bind more strongly compared to its monomeric counterpart (Figure 4). Indeed, in order for the monomeric aptamer to dissociate from the target, only one binding interaction must be broken, while for the multimeric aptamer, all binding interactions must be disrupted simultaneously. This principle is referred to as avidity. 

Following this concept, bivalent or multivalent aptamer constructs have been synthesized, which are composed of at least two identical or different aptamer sequences that are joined together via a covalent linker or via non-covalent interactions. By correct choice of linker type, length and flexibility, the binding strength of the multivalent aptamer can largely exceed that of the monovalent counterpart [75,76].

Easley et al. developed a bivalent aptamer probe for the detection and quantification of the protein thrombin using thermofluorimetric analysis [77]. The aptamer probe consists of two thrombin binding aptamers which are elongated with a tail sequence. Afterwards, the two tail sequences are non-covalently conjugated via the hybridization with a paired bivalent probe, which allowed sensitive detection. The concept of avidity is also of particular interest in the field of biosensors, where sensor surfaces are loaded with multiple copies of the aptamers. In 2018, Shao et al. reported on the influence of aptamer surface density on the binding efficiency for the inactivated H1N1 virus [74]. Surfaces with high aptamer density were significantly more sensitive towards the virus, which lowered the limit of detection and increased the application potential of the aptasensor.

The enhanced binding efficiency of multivalent aptamers has also been employed for the capture of cells. This is especially valuable in the field of cancer research, where isolation of circulating tumor cells remains challenging. Fan et al. reported on the conjugation of up to 95 aptamers on a single gold nanoparticle, which increased the binding affinity towards leukemia cells with a factor of 39 compared to the single aptamer [78]. This approach shows great potential for the sensitive detection of tumor cells in cancer diagnosis. Another strategy, developed by Zhang et al., made use of the NanoOctopus construct. In this case, thousands of aptamers targeting the tyrosine kinase 7 (PTK7) protein were conjugated via T20 linkers. The resulting multimeric aptamer-tentacles were subsequently immobilized on magnetic nanoparticles, which allowed the efficient and selective isolation of PTK7 overexpressing CCRF-CEM cells [79]. More recently, the research group of Xian et al. developed a method for isolating and detecting circulating tumor cells in blood samples by using tetravalent DNA structures [80]. The overall construct contained aptamer sequences for the recognition of MUC-1 (a surface protein overexpressed on various cancer cells), a biotin moiety for efficient isolation and Ag_2_S nanodots for near-infrared fluorescent detection. The capture efficiency and purity of the isolated circulating tumor cells were 98% and 97% respectively, thus, demonstrating the great potential of this methodology for cancer detection.

#### 3.1.3. Exploiting the Extended Genetic Alphabet: Artificial Nucleobase Incorporation into Aptamers

Expanding the genetic code by means of unnatural base pairs is another strategy for generating customized nucleic acid libraries with increased functionality and variability. Until now, different unnatural base pairs have been designed for diverse purposes [81] and throughout the years, these building blocks have been optimized for high fidelity incorporation by polymerases, in PCR amplification and sequencing efforts [82]. Herein, we will solely focus on the dDs-dPx (Ds: 7-(2-thienyl)-imidazo[4,5-b] pyridine, Px: 2-nitro-4-propynylpyrrole) and the dZ-dP (dZ: 6-amino-5-nitro-3-(1′-β-D-2′-deoxyribofuranosyl)-2(1H)-pyridone, dP: 2-amino-8-(1′-β-D-2′-deoxy-ribofuranosyl)-imidazo[1,2-a]-1,3,5-triazin-4(8H)-one) base pairs (Table 1), as these have been implemented during the development of specific diagnostic tools such as ELONA assays and aptasensors (Figure 5 and Figure 6). Both bases (dDs and dZ) recognize their complementary unnatural base (dPx and dP, respectively) in different ways. The dDs-dPx base pair complementarity is mediated by hydrophobic and packing interactions, and thereby, exploits the use of shape complementarity of the unnatural bases (Figure 5). The large thienyl moiety of dDs successfully prevents interaction with natural bases, while the nitro-group of dPx allows avoiding of the potential interaction with natural adenine due to electrostatic repulsion between the nitro-group and the N1 nitrogen of adenine (which is absent in the dDs complement). The propynyl group of dPx was introduced to increase the hydrophobicity of the nucleobase, and therefore, increases the affinity for polymerases [82]. In contrast, the dZ-dP pair uses the classical hydrogen bonding complementarity (the purine base of dP interacts via an acceptor–acceptor–donor pattern with the pyrimidine base of dZ) (Figure 6). This interaction pattern is orthogonal to that of natural nucleobases, thereby preventing mismatch formation [83]. By using genetic alphabet Expansion SELEX, a variation of the conventional SELEX that uses a random library of sequences modified with 1–3 dDs modifications on fixed positions, dDs modified aptamers against the vascular endothelial growth factor VEGF-165 [84], the interferon-γ cytokine (IFN-γ) [84] and the Von Willebrand Factor (VWF) [85] were selected, the former two showing more than a 100-fold increase in binding affinity (pM range) compared to the aptamer sequences containing solely natural nucleobases. Selected dDs-modified aptamers were examined for their applicability in ELONA assays [86]. Different designs were explored in order to identify the most efficient set-up. Immobilization of the target proteins (VEGF-165 or INF-γ) on a microtiter well plate was combined in a competitive design with biotinylated dDs aptamers (for recognition) and streptavidin horseradish peroxidase (HRP) (for detection) (design A, Figure 5). The second and third designs were based on a sandwich assay, in which monoclonal antibodies (mAb) and dDs aptamers were interchanged as detection or capture agents (designs B and C, Figure 5). When comparing the different strategies, it was observed that the sensitivity of the sandwich assays exceeded that of the competitive design. The difference in sensitivity may also be the result of varying surface immobilization efficiencies between proteins (design A), antibodies (design B) and aptamers (design C). Interestingly, aptamers with higher affinity for their target showed a higher sensitivity and signal intensity, which greatly favors the applicability of dDs-modified aptamers over regular aptamers for the development of improved diagnostic devices.

Recently, Tan et al. intertwined the advantages of dZ-dP-modified aptamers, electrochemical sensing and DNA nanostructures for the detection of the hepatocellular HepG2 exosomes [87]. Exosomes mediate intercellular communication and are identified as early cancer biomarkers. By immobilizing DNA nanotetrahedrons bearing the dZ/dP aptamer on a gold surface, they circumvented the formation of self-assembled monolayer aggregates frequently observed when immobilizing the mere aptamer due to the flexibility of the single stranded DNA. This modified set-up allowed the efficient detection of hepatocellular cancer exosomes (Figure 6) with a 100-fold enhanced selectivity compared to single stranded aptamers. The technology shows potential for the development of future detection assays targeting disease biomarkers in complex body fluids. Other applications of aptamers containing unnatural nucleobases can be found in the development of nested PCR assays for the multiplexed detection and quantification of viral DNA/RNA [88,89]. As the world of artificial nucleobases is still emerging and evolving at a rapid pace, as recently demonstrated by Hachimoji eight-letter DNA/RNA [90] and the design of semi-synthetic organisms [91], we foresee, in the near future, an increase in diagnostic applications which exploit the advantages of aptamers with an extended genetic alphabet.

#### 3.1.4. Nucleotides with Amino Acid Like Side Chains

To meet the problem of limited functional diversity in nucleic acids, nucleobases can be decorated with a wide variety of functional groups. A clever strategy proposed to combine the best characteristics of both nucleic acids and proteins involves decorating the nucleobases with amino acid-like side chains. In general, two different strategies have been exploited where the amino acid side chain is either (1) directly involved in the binding interactions with the target or (2) involved in stabilizing the aptamer structure, which can indirectly increase the binding properties.

By decorating the C5-position of deoxyuridine triphosphates (dUTP) with a benzyl- (BndU), naphthyl- (NapdU), tryptamino- (TrpdU) or isobutyl derivative (iBudU), Gold et al. succeeded in combining the conformational flexibility of nucleic acids (responsible for the unique ability of aptamers to bind with their target), with the functional diversity of proteins (Figure 7) [35,92]. These modifications greatly enhance the binding affinity of the corresponding aptamers towards proteins. The structural basis behind the increased binding strength lies in the formation of additional hydrophobic interactions between the dUTP modification at C5 and the hydrophobic pockets of the proteins [35,93]. These aptamers were named SOMAmers (Slow Off-rate Modified Aptamers), which refers to the stringent selection criteria used for their isolation. Indeed, only high affinity aptamers with slow dissociation rates (t_1/2_ > 30min) were selected from the initial DNA library. Gawande et al. further expanded this concept and developed ‘next-generation’ SOMAmers, which are characterized by modifications at the C5 position of two pyrimidine bases (deoxycytidine triphosphate dCTP and deoxyuridine triphosphate dUTP) [94,95]. A total of two modifications on dCTP (naphthyl or phenyl) and five modifications on dUTP (naphthyl, phenyl, tyrosine, morpholino or threonine) were combined and used as a starting point for SELEX selection against the human model protein PCSK9 (Figure 7) [94,95]. Hydrophobic modifications on both dCTP (NapdC and PpdC) and dUTP (TyrdU) increase the binding affinity, with K_D_ values in the pM range. The enhanced binding strength and specificity observed with double modified SOMAmers emphasize the importance of hydrophobic interactions for the selective binding of proteins. The unique characteristics of single or double modified SOMAmers makes them ideal candidates for diagnostic assays (vide infra). Thus, further diversification of SOMAmers via post-SELEX strategies may be a valuable tool for improving their applicability and market potential.

As the proteome is highly dynamic in nature and quickly responds to external and internal changes, proteins are powerful biomarkers for monitoring human health. Unraveling the language of proteins translates to a better understanding and earlier prognosis of diseases. Moreover, a better knowledge of the proteome paves the way for targeted personalized healthcare. Nowadays, a wide variety of antibody-based assays exist for the sensitive detection of protein biomarkers. However, the cross-reactivity of antibodies is a common problem [96]. Furthermore, the high cost and poor reproducibility of antibody synthesis is hampering the development of affordable assays, which prompts researchers to look for better alternatives [97]. In the year 2000, Larry Gold founded the company SOMALogic which developed the SOMAscan platform (Figure 8) [98]. In this DNA microarray platform, 5′-biotinylated SOMAmers are modified with a photocleavable linker and a fluorescent tag followed by coating on streptavidin beads. The obtained functionalized beads are incubated with the sample (e.g., blood, plasma, serum), allowing cognate and non-cognate interactions between the SOMAmers and the proteins (Figure 8, step a). Subsequently, unbound proteins are removed from the beads via a washing step (step b), followed by biotinylation of the bound proteins (step c). Next, the SOMAmer–protein complexes are released from the beads via UV irradiation of the photocleavable linker (step d) and non-specific interactions are disrupted using a polyanionic competitor (step e). SOMAmers are like any other single stranded DNA molecule—polyanionic. Selective disruption of non-specific SOMAmer–protein interactions by a polyanionic competitor is based on the difference in dissociation rate between strongly bound target molecules and weakly bound off-target proteins. Next, the target protein–SOMAmer complexes are recaptured on a new set of streptavidin-coated beads (step f), thereby further increasing the specificity of the process. Finally, the surface bound protein–SOMAmer complex is disrupted under denaturing conditions (step g) and the released SOMAmers are hybridized to their complementary immobilized sequences in a microarray format (step h). The subsequent fluorescent read-out gives a direct quantification of the amount of protein present in the sample, as the SOMAmer forms a 1:1 complex with its target protein. This assay highlights two key features of SOMAmers: (1) their dual ability to bind proteins and hybridize with their complementary nucleic acid strand and (2) their ability to refold after denaturing conditions [99]. Hence, the SOMAscan platform is a powerful alternative to antibody assays, which allows a high degree of multiplexing (more than 1000 proteins measured simultaneously) with a sensitivity comparable to ELISA-type assays [98]. As such, this platform is ideal for studying the proteome in health and disease. The pertinence of the SOMAscan technology is exemplified by the variety of applications describing its use and can hardly be overestimated. The SOMAscan assay has been employed for the detection of cancer [100,101,102,103,104], lung- [105,106], neurological- [107,108], renal- [92], inflammatory- and infectious disease biomarkers [109,110,111,112,113,114,115]. Other health-related problems such as cardiotoxicity [116], metabolic dysregulation [117] and aging [118] have also been investigated with the technology. Moreover, the assay can be applied for comparative biomarker studies [119] or for monitoring deviations in plasma protein constitution during pregnancy for the early detection of abnormalities in fetal development [120].

In addition to their use in the SOMAscan platform, SOMAmers have been employed in sandwich-type diagnostic assays. One documented strategy is the combination of antibodies and SOMAmers for the detection of toxin A, B and binary toxins, which are indicators of *Clostridium difficile* infections. Both well plate-based and membrane-based assays were tested [121]. In the former, biotinylated SOMAmers were immobilized on a streptavidin plate and used as capture agents. Addition of a primary monoclonal antibody against the toxins resulted in the formation of a sandwich complex, which was transduced to a measurable signal using a secondary antibody horseradish peroxidase conjugate. In the membrane-based assay, the monoclonal antibody was used as the capture reagent, whereas the biotinylated SOMAmer was employed for sandwich complex formation. The subsequent addition of streptavidin-alkaline phosphatase allowed quantification of the toxins. Both methods showed comparable results with higher detection limits observed for the membrane-based assay compared to the well plate format. A similar strategy was applied in single molecule arrays (Simoa array) for the detection of the tumor necrosis factor α (TNF-α) cytokine. In this sandwich assay, a capture antibody was immobilized on paramagnetic microbeads while sandwich formation was induced through the addition of biotinylated SOMAmer reagents. Fluorescent detection was performed using a streptavidin β-galactosidase, binding with the biotinylated SOMAmer, which hydrolyses non-fluorescent resorufin-β-d-galactopyranoside to fluorescent resorufin [97].

An alternative and antibody-free strategy based on SOMAmers was developed for the detection of the binary toxin from *Clostridium difficile* (the so-called Luminex platform) [122]. Herein, non-compatible SOMAmer pairs were selected for the binding of different toxin epitopes. All SOMAmers were labeled with biotin and served either as capture probes or detecting agents. Capture probes were immobilized on beads, while the detecting probes were added after target binding. Upon binding of the latter with the protein target, the sandwich complex could be detected using streptavidin-HRP. A signal is only detected upon sandwich complex formation, which can only take place in the presence of the toxin. The sandwich SOMAmer assay was also used for the discrimination and detection of the two protein isoforms, GDP-8 and GDP-11 [123]. In a similar approach, a SOMAmer named HVP-07 was selected against the Type 16 virus-like particle (VLP) and used in a sandwich-like mix and read assay. The use of SOMAmers in this assay greatly improved the simplicity of the system (no wash steps required) and allowed for a fast read-out [124].

While the majority of applications are performed with single base modified SOMAmers, as is the case for the SOMAscan platform or sandwich assays, the possible modifications and applications of SOMAmers reach far beyond this scope. For example, sandwich assays are described with double base-modified SOMAmers [95] and SOMAmers can be used for on-chip electrophoretic assays [125], the development of gravimetric aptasensors [126] or even for affinity pull-down experiments [127], highlighting the market potential of aptamers containing modified nucleobases.

Alternatively, besides the predominantly hydrophobic moieties that have been introduced (vide supra), highly hydrophilic residues can also be incorporated, which can result in additional H-bonding and ion–ion interactions. Over the years, our research group has gathered ample expertise in the design and incorporation of the imidazole-tethered thymidine (T^ImH+^) in synthetic DNA duplexes (Figure 9A) [128]. Initial exploration has resulted in the discovery of a ‘stabilizing motif’ (Figure 9B), which has proven to significantly increase the thermal stability of a DNA duplex. The increase in melting temperature could be attributed to a stabilizing hydrogen bond between the imidazole and the carbonyl oxygen (O6) of the *n*+2 guanine of the opposing strand. In a proof-of-principle study, the T^ImH+^ building block was incorporated in the stem region of the L-argininamide binding aptamer, resulting in enhanced stability and potential for the design of more stable aptamers, while retaining binding affinity for the target [129]. This methodology is currently further investigated by our research group for determining the correlation between the stabilization of the aptamer in its binding competent state and the potential increase in binding affinity for its use in aptasensors.

Ongoing research, which is focusing on the synthesis and incorporation of a large variety of modified nucleosides, will eventually open the doors to a further extension of the structural diversity of nucleic acid structures with amino acid-like side chains [130,131]. We foresee that the obvious potential of the latter will drastically increase the applicability of aptamers in further diagnostic applications [132].

#### 3.1.5. 2′-Fluoro Arabino Nucleic Acid (2′F-ANA)

2′-Fluoro arabino nucleic acid (2′F-ANA, Figure 10A) is a fluoro-derivative of arabinose which has successfully been implemented as a nucleotide-mimic in non-canonical structures such as G-quadruplexes (Figure 10B) [133]. These stable secondary structures are formed in guanine-rich RNA or DNA oligonucleotides. The structure of G-quadruplexes can be attributed to the planar stacking of guanine tetrads, in which four guanines interact via Hoogsteen hydrogen bonding.

The impact of 2′F-ANA substitution in G-quadruplexes has been thoroughly investigated for the thrombin binding aptamer (TBA) [134]. Substitution of anti-positioned guanines with their 2′F-ANA counterpart results in an increase in thermal and structural stability together with a 4-fold increase in binding affinity towards thrombin. Conversely, when syn-oriented deoxyguanosines were substituted, a conformational switch is often observed, which had a detrimental effect on the stability and binding ability of the aptamer. In a follow-up study, mapping of the affinity landscape of 2′F-ANA-modified TBA aptamers was undertaken [135]. In total, 32,768 sequences were synthesized on a microarray in order to map all possible 2′F-ANA substitutions of the aptamer. Afterwards, thrombin was added at various concentrations followed by the addition of a second fluorescently-labeled TBA aptamer for determination of the binding constant. From this experiment, it was concluded that syn-orientated guanines do not tolerate modifications, while substitution of anti-oriented guanines for 2′F-ANA is indeed possible. Finally, it was noticed that incorporation of 2′F-ANA in the two TT loops of the aptamer had a beneficial effect on the binding affinity.

Recently, 2′F-ANA-modified aptamers have been identified via SELEX which possess affinity for the human integrase enzyme of the human immunodeficiency virus (HIV). This enzyme is responsible for the integration of viral DNA into the genome of the host and is crucial for the infection cycle of the virus [136]. The two 2′F-ANA-modified aptamers obtained after the selection process have dissociation constants in the 50–100 pM range, which is more than two orders of magnitude stronger compared to previously reported unmodified DNA and RNA aptamers. Substituting the 2′F-ANA nucleotides with their fluorinated RNA analogues drastically reduced the binding affinity, thereby suggesting that the 2′F-ANA nucleotides are inducing a unique conformation in the aptamer, which is crucial for high affinity binding to the integrase enzyme. While the current applications of 2′F-ANA-modified aptamers are limited, the promising results reported so far warrant further investigation. In doing so, the full aptamer enhancing potential of these unnatural nucleotides may be unveiled.

#### 3.1.6. Mono- and Dithioaptamers

Chemical modification of aptamers can also be envisioned on the phosphodiester linkage instead of at the nucleobase level. Thioaptamers are aptamer derivatives, where one (monothio) or both (dithio) non-bridging phosphodiester oxygens are substituted with sulfur (Figure 11). In the case of monothioaptamers, a chiral center is generated on the phosphor atoms during synthesis, with concomitant formation of diastereoisomers. For dithioaptamers, the backbone phosphor atoms are achiral. Replacing one or more oxygen atoms with sulfur results in altered characteristics of thioaptamers compared to regular oligonucleotides, such as increased nuclease resistance and stronger binding towards protein targets [137]. The latter is linked to the soft anion character of sulfur (compared to the hard anion character of oxygen), which results in a less favorable soft–hard interaction between the sulfur anion and hard cationic counterions surrounding the phosphate backbone, like sodium or potassium [138]. Consequently, upon protein binding, less energy is needed to strip away these counterions from the sulfur anions compared to oxygen, which, in turn, results in the observed enhanced binding affinity of thioaptamers [21]. However, care must be taken when considering these backbone modifications, as a too high substitution ratio of oxygen atoms with sulfur results in non-specific binding interactions, hence, emphasizing the need for selection methods, where the number and position of thiosubstitutions can be controlled. For this purpose, combinatorial- [139], bead-based- [140] and high-throughput [141] selection methods have been described.

To date, mono- and dithioaptamers have been selected against multiple pharmaceutical targets, such as the transcription factors NF-IL6 [139] and NF-κB [142], the reverse transcriptase of HIV [143], the transforming growth factor TGF-β1 [144] and the receptor activator of NF-κB (RANK) [145]. Thioaptamers were also used in an on-chip capture and digestion protocol to identify unknown proteins in biological samples. In this procedure, called the SELDI-MS assay (surface enhanced laser desorption ionization MS), the amino-modified thioaptamer XBY-S2 was immobilized on a P20 ProteinChip Array, followed by incubation with nuclear extracts derived from lipopolysaccharide-stimulated mouse 70Z/3 pre-B cells nuclear extracts. After washing and trypsin digestion, the proteins captured by the aptamers could be identified via tandem-MS. In this way, five mouse heterogeneous nuclear ribonucleoproteins were identified. Verification of their identity was conducted with a thioaptamer/antibody sandwich assay [146]. A second application of thioaptamers is the development of an electrochemical biosensor for the detection of TGF-β1, released from stellate cells. TGF-β1 are proteins associated with fibrosis of the liver and other organs. In the liver, these proteins are secreted by activated stellate cells. In the assay, an amino-modified aptamer targeting TGF-β1 was conjugated to methylene blue as a redox label and immobilized on gold electrodes. This set-up was integrated in a microfluidic device. Upon activation of the stellate cells, TGF-β1 is secreted and binds to the thioaptamers on the electrode surface, which generates a measurable electrochemical signal proportional to the TGF-β1 concentration and secretion rate [147]. As a final example, a thioaptamer against NF-κB was utilized for the development of an integrated electrophoretic on-chip gel-shift detection assay [148].

#### 3.1.7. Locked Nucleic Acid (LNA) and Their Potential Usefulness for Diagnostic Tool Development

A final unnatural nucleoside that will be discussed is the locked nucleic acid (LNA). This analogue differs from regular nucleosides as its conformational structure is ‘locked’ via a methylene bridge linking the 2′ oxygen and the 4′ carbon atoms of the sugar ring, thereby fixing the nucleoside in a C3′-endo conformation (Figure 12) [149]. The rigidity of the system enforced by this modification accounts for the high thermal stability and nuclease resistance of LNA-modified aptamers, which increases their potential in therapeutic and diagnostic applications. While successful combination of both pre-SELEX (LNA aptamer selection) and post-SELEX (truncation) methodology has been reported in the literature, diagnostic tools incorporating LNA-modified aptamers have not been widely applied yet [150]. Nevertheless, it is clear from what follows that LNA-modified aptamers offer promising opportunities in this context. 

The first study employing LNA-modified aptamers was conducted on the G-quadruplex-forming thrombin aptamer [151]. LNA was introduced at varying positions to assess its influence on biological activity. While the introduction of the LNA moiety in all cases increased the thermal stability of the aptamer, the same could not be said for binding affinity. Depending on the position, LNA aptamers showed a decreased or comparable biological affinity compared to the unmodified aptamer. Since this report, multiple aptamers such as the (strept)avidin binding aptamer [152,153], the Sgc8 aptamer [154] and the TD05 aptamer [155] have been modified with LNA, with a similar result in all cases. In general, the structural changes and biological effects that LNA modifications impose are dependent on the position and number of modifications. When focusing solely on binding affinity, modifications are not well tolerated in regions crucial for target binding, as this often results in a decrease or complete loss of binding. Moreover, aptamers undergoing large structural changes upon target binding often tolerate LNA modifications to a lesser extent. Hence, introducing LNA modifications in aptamers is a delicate process that requires time, trial-and-error and fine-tuning. However, when done correctly, the properties of these aptamers can transcend those of regular aptamers in terms of thermal and structural stability, nuclease resistance and binding affinity.

A potential diagnostic application encompasses the use of LNA aptamers as fluorescent biosensors such as molecular beacons. The use of aptamers as molecular beacons has been previously described (reviewed in [156]), but not yet in combination with LNA modifications. An example of a molecular beacon is the thrombin-binding aptamer. This aptamer beacon can be modified with a quencher and fluorophore at the 3′ and 5′ end (Figure 13). The complementarity of the 3′ and 5′ end forces the aptamer in a stem-loop structure. In this conformation, the fluorophore is efficiently quenched. Upon binding of the thrombin, the aptamer adopts a G-quadruplex structure, creating a distance between the fluorophore and quencher and results in the subsequent release of a fluorescent signal [157]. Designing such systems while incorporating LNA building blocks could be beneficial since (1) correct placement of LNA could stabilize G-quadruplex formation [158] and (2) the thermal and nuclease stability of the system would be greatly enhanced, allowing the use of more stringent conditions during the analysis of complex biological samples. We envision that the applicability of LNA aptamers will reach beyond fluorescent biosensors or G-quadruplex forming aptamers. Since LNA-modified aptamers combine the benefits of LNA modification with the remarkable selectivity of aptamers, they can potentially be used for the development of highly multiplexed arrays for the fast detection of e.g., disease biomarkers.

## 4. Modified Aptamers for Covalent Target Trapping

The examples discussed so far employed aptamers obtained via the SELEX process. Hence, the aptamers were selected against a known (protein) target. This is in contrast to the cell-SELEX methodology, in which aptamers are selected for binding whole cells. Despite the great potential of cell-SELEX, the unknown identity of the target requires subsequent target identification, which is a major limitation of the technology. Unfortunately, the use of unmodified aptamers for target identification purposes is rather restricted due to the non-covalent nature of the aptamer–protein interactions [159]. To solve this problem, a second strategy was developed for enhancing the binding affinity, which uses a covalent linkage between the aptamer and its target, hereby allowing robust pull-down of the complex and its subsequent identification. Over the years, a few UV activatable crosslinkers have been evaluated for this purpose, which are mostly incorporated post-SELEX.

### 4.1. 5-Iodo-2′-Deoxyuridine (5-IdU)

In 2007, the research group of Tan et al. successfully identified an aptamer that recognized a membrane-bound protein on Raman cells [160]. In order to identify the unknown target, the 3′ biotinylated aptamer was modified with the photoreactive 5-iodo-2′-deoxyuridine moiety (5-IdU, Figure 14A). Upon UV irradiation, a radical intermediate is generated which can form a covalent crosslink with the protein target. After successful crosslinking, pull-down of the complex is performed using the biotin handle, followed by identification of the target. Studies indicated that the degree of thymine substitution for 5-IdU should be monitored with care, since fully modified aptamers were not capable of recognizing the protein. Despite the potential of this methodology, results with other aptamer sequences were not satisfying. Therefore, new and versatile photoreactive modifications are required, which are more broadly applicable.

### 4.2. Phenyl Azide-Modified Aptamers

In an attempt to provide a suitable alternative to the 5-IdU modification, the photoreactive phenyl azide-based crosslinking approach was developed [161]. This methodology uses a modified nucleoside building block, which contains both a phenyl azide and a biotin moiety (Figure 14B) [161]. This building block is attached to the 5′-end of the aptamer to prevent disruption of the binding interaction, which was confirmed with a filter retention assay. The pull-down strategy of this technique is similar to the example given above for 5-IdU. After mixing the phenyl azide-modified aptamer with the unknown target, UV irradiation of the sample is performed. The generated reactive nitrene intermediate on the aptamer reacts with the target with concomitant formation of a covalent crosslink. Once formed, pull-down of the complex with streptavidin-coated magnetic beads and subsequent identification of the target is performed. To evaluate the overall applicability of the methodology, three structurally diverse aptamers were investigated possessing secondary structures ranging from G-quadruplexes, hairpin loops and double bulges. Additionally, the three protein targets were located in different environments: on the cell-surface, in the cytoplasm or extracellularly in the blood stream. After UV irradiation, crosslinking was observed which indicates that the phenyl azide moiety was in close proximity to the protein. The strategy was successfully implemented for the efficient pull-down of target proteins, even in highly complex biological samples. The broader applicability of phenyl azide compared to 5-IdU-modified aptamers, makes them ideal tools for efficient target labeling and identification. 

### 4.3. Novel Crosslinking Aptamers with Future Potential for Diagnostic Tool Development

Progress in triggerable and non-triggerable crosslinking approaches is evolving rapidly and new crosslinking moieties have been incorporated into aptamers for the purpose of covalent target binding. While strictu sensu, the technologies discussed below have not been applied yet in a specific diagnostic tool setting but have only been investigated from a more fundamental point of view, we believe that their promising potential justifies including them in the discussion.

#### 4.3.1. Diazirine-Modified Aptamers

An alternative photoreactive crosslinking aptamer for efficient labeling and pull-down of unknown targets is the diazirine-modified aptamer shown in Figure 15. In 2014, Yang et al. described the synthesis of two diazirine-modified aptamers, which bind to streptavidin and thrombin [162]. First, the effect of incorporating the modified diazirine building block at different locations in the aptamers sequence was investigated. It was clear that introducing the diazirine moiety on certain positions had a major impact on target binding, thus, demonstrating the importance of fine-tuning the positioning of modifications during the aptamer optimization process. Having identified suitable diazirine-modified aptamers with retention of binding capability, both aptamers were employed for the pull-down of streptavidin and thrombin. UV irradiation of the diazirine moiety results in the formation of a reactive carbene intermediate responsible for crosslinking to the target. A high specificity and pull-down efficiency was observed for both aptamers, which highlights their potential in the identification of unknown protein targets.

A slightly more complex design was adopted by Zhang et al. for the successful aptamer-mediated pull-down of the LYS protein in complex media [159]. This methodology makes use of an aptamer that acts as a binding probe for a biotin- and diazirine-modified capture probe. However, this method is an aptamer-mediated pull-down approach, since the aptamer is used to migrate the actual capture probe towards the target for induced proximity between the protein and the crosslink agent. Due to this fact, this approach is not discussed in more detail in this review. This elegant strategy demonstrates the versatility of aptamers and the potential of a diazirine-mediated crosslinking agent for target pull-down efforts, in complex biological systems.

#### 4.3.2. α, α-Gem-Difluoromethyl Carboxylic Acid (F-carboxyl) and Aldehyde-Modified Aptamers

Protein–aptamer directed ligation is a method that introduces a covalent linkage between an oligonucleotide and a protein target. In a template-directed set-up, a protein–oligonucleotide conjugate is formed. Herein, an aptamer functions as template or recognition unit, bringing a protein and a reactive oligonucleotide in vicinity, thereby directing the selective distance-dependent reaction to occur (Figure 16A). In a directed non-templated set-up, the aptamer itself is modified with a reactive group. Binding of the aptamer to its target brings the reactive moiety on the aptamer in close proximity to the target with concomitant formation of a covalent link (Figure 16B). In this section, our area of focus will be the directed reactions as they require modification of the aptamer with a crosslink moiety. These ligation reactions differ from the previous discussed modifications, as the incorporation of the reactive group does not enhance the binding affinity (K_D_ value) of the aptamer. Nevertheless, once a covalent linkage is formed, dissociation of the aptamer from its target is not possible anymore (under standard conditions), which renders the K_D_ theoretically ‘infinite’. We will discuss two types of directed ligation methods employing an F-carboxyl (α, α-gem-difluoromethyl carboxylic acid) or an aldehyde as the reactive group for which we foresee potential in diagnostic applications. Previously discussed crosslink strategies relied on the generation of reactive species by UV irradiation. In contrast, F-carboxyl- or aldehyde-modified aptamers do not require photoactivation.

The first application of F-carboxyl-modified aptamers is based on their ability to site-specifically and selectively conjugate cell surface proteins [163]. The F-carboxyl moiety is known to have moderate reactivity towards amino functionalities. Upon binding of the modified aptamer with its protein target, the F-carboxyl on the aptamer is brought in the vicinity of amino functionalities present on the protein (Figure 17A). This allows the proximity-induced reaction to take place. Protein–aptamer directed ligation has also been performed employing aldehyde-modified aptamers. The latter can be synthesized starting from the corresponding vicinal diol, using NaIO_4_ oxidation. After binding of the aptamer to its target, reaction of the aldehyde moiety with a nearby lysine of the protein is induced. While the formation of the Schiff base is reversible, reduction with NaBH_3_CN results in the formation of a stable covalent bond between aptamer and protein (Figure 17B) [163]. 

Both strategies could be useful in a diagnostic context e.g., for the detection of protein biomarkers in multiplexed assays. Specifically, the introduction of a covalent linkage between the aptamer and its target would be beneficial, as more stringent washing conditions could be employed in order to remove non-specific binders. Recently, an aldehyde-modified aptamer was used for the covalent trapping of a therapeutic human antibody of the IgG1 type. In this work, the aldehyde functionality was incorporated in the hairpin region of the aptamer (not involved in target binding) through the reaction of an amino-modified aptamer and an aldehyde NHS-ester. Upon target binding, the aldehyde reacts with a nearby lysine residue on the antibody. The formation of a covalent bond is guaranteed through the reduction in the Schiff base. This direct ligation strategy could be a valuable tool in the development of detection methods such as immuno-PCR and ELISA [164].

## 5. Conclusions

Aptamers have proven as powerful substitutes for antibodies in diagnostic applications due to their exceptional recognition properties towards a variety of associated targets. However, the need arises for improved ‘second-generation’ aptamers due to their confined structural diversity and stability. To this end, multiple unnatural nucleotides have been developed which, after incorporation into the aptamer sequence, have the potential to enhance the aptamer properties. In this review, we focused on modifications, introduced in a pre- or post-SELEX manner, that directly or indirectly influence the binding event of the aptamer, resulting in enhanced target binding. Moreover, we elaborated on their utility and proved their potential in diagnostic assays.

It can be noted that modified aptamers as diagnostic agents have not yet been used to their full extent. Despite the impressive research and progression that has been made in this field, there is still scope for greater refinements, and thus, more successful future exploitation of their full potential as truly versatile reagents. Indeed, there is a need for cheaper diagnostic devices containing stable aptamers with high affinity for their targets. The modifications reported in this review may provide the means to achieve this goal.

## Figures and Tables

**Figure 1 ijms-21-04522-f001:**
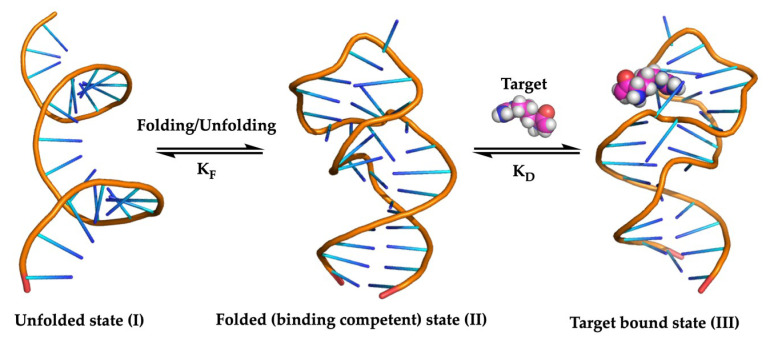
Equilibrium between the unfolded aptamer state (**I**) and its binding competent state (**II**) with corresponding equilibrium constant K_F_. The binding competent state (**II**) is able to bind the target (conformation **III**), which is characterized by the dissociation constant K_D_.

**Figure 2 ijms-21-04522-f002:**
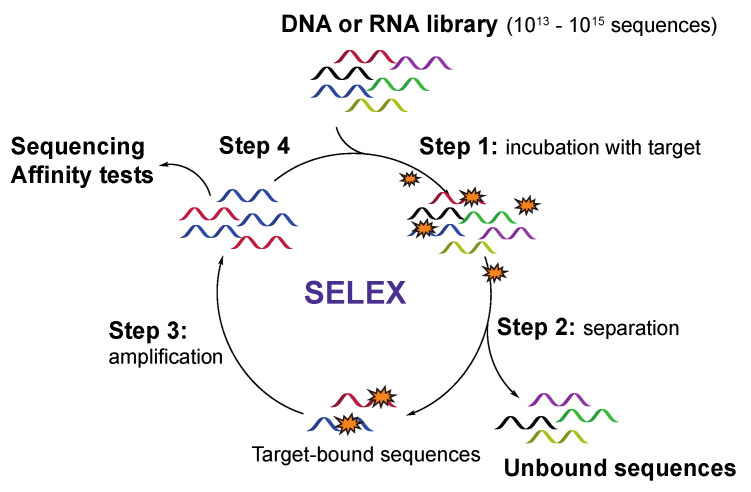
Schematic representation of a conventional SELEX process for the selection of aptamers with high affinity against a target of interest.

**Figure 3 ijms-21-04522-f003:**
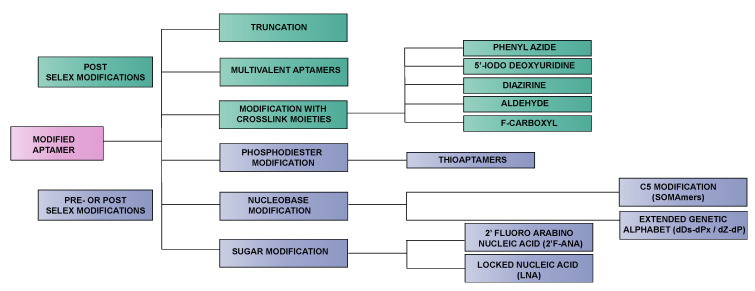
Flow diagram visualizing the aptamer modifications discussed in this review. These modifications are incorporated either via a pre- and/or post-SELEX strategy in the aptamer sequence. The modifications in the green boxes are typical post-SELEX modifications, while the modifications in the purple boxes can be introduced both before (pre) or after (post) the SELEX process.

**Figure 4 ijms-21-04522-f004:**
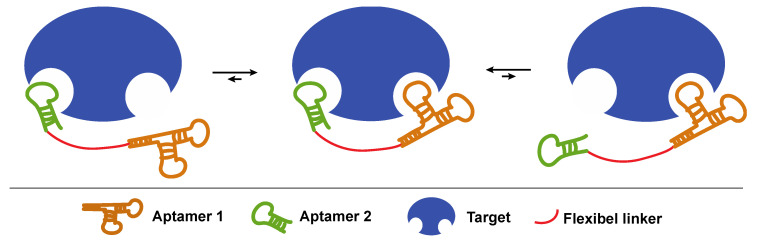
Principle of avidity explained for a bivalent aptamer towards a multimeric target. The dimeric aptamer construct consists of two aptamers conjugated by a flexible linker.

**Figure 5 ijms-21-04522-f005:**
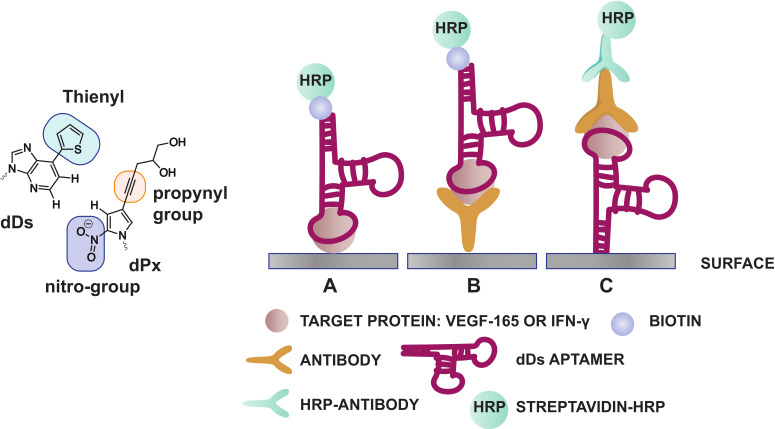
Structure of the dDs and dPx unnatural base pairs (**left**). Illustration of competitive (**A**) and sandwich ELONA assays for the recognition of the proteins VEGF-165 and INF-γ using immobilized antibody (**B**) and immobilized dDs-modified aptamer (**C**) (**right**).

**Figure 6 ijms-21-04522-f006:**
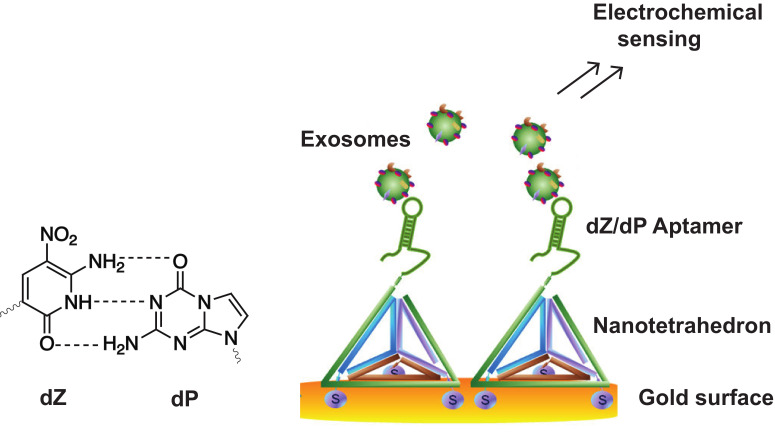
Structure of the dZ-dP unnatural base pair and the corresponding hydrogen bonding interactions (left). Aptasensor for the electrochemical detection of HepG2 exosomes using nanotetrahedrons bearing the dZ-dP aptamers immobilized on a gold surface (right). Reprinted (adapted) with permission from ACS Nano 2017, 11, 4, 3943–3949. Copyright (2017) American Chemical Society.

**Figure 7 ijms-21-04522-f007:**
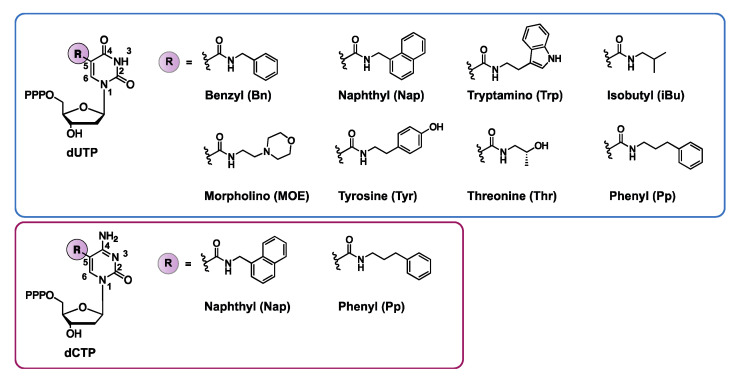
Overview of nucleobase modifications introduced at the C5 position of dUTP (deoxyuridines triphosphate) or dCTP (deoxycytidine triphosphate) for the synthesis of so-called SOMAmers. PPP—triphosphate. 5-benzylaminocarbonyl (Bn), 5-naphthylmethylaminocarbonyl (Nap), 5-tryptaminocarbonyl (Trp), 5-isobutylaminocarbonyl (iBu), 5-morpholinoethylamino-carbonyl (MOE), 5-hydroxyphenylethylaminocarboxyl (Tyr), 5-hydroxypropylaminocarbonyl (Thr), 5-phenylpropylaminocarbonyl (Pp).

**Figure 8 ijms-21-04522-f008:**
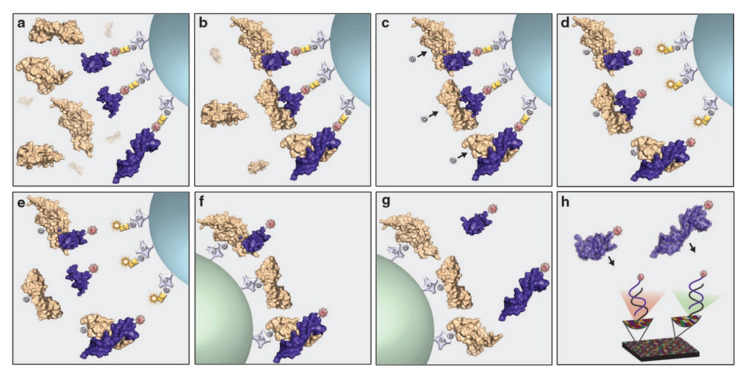
Schematic representation of the SOMAscan assay. Beige structure—protein in sample, purple structure—SOMAmer with affinity for target protein, grey circle B—biotin on streptavidin (SA)-coated bead, yellow square L—photocleavable linker, red circle F—fluorophore. In this DNA microarray platform, 5′-biotinylated SOMAmers are modified with a photocleavable linker and a fluorescent tag followed by coating on streptavidin beads. The obtained functionalized beads are incubated with the sample (e.g., blood, plasma, serum) allowing cognate and non-cognate interactions between the SOMAmers and the proteins (step **a**). Subsequently, unbound proteins are removed from the beads via a washing step (step **b**) followed by biotinylation of the bound proteins (step **c**). Next, the SOMAmer–protein complexes are released from the beads via UV irradiation of the photocleavable linker (step **d**) and non-specific interactions are disrupted using a polyanionic competitor (step **e**). Next, the target protein–SOMAmer complexes are recaptured on a new set of streptavidin-coated beads (step **f**), thereby further increasing the specificity of the process. Finally, the surface bound protein–SOMAmer complex is disrupted under denaturing conditions (step **g**) and the released SOMAmers are hybridized to their complementary immobilized sequences in a microarray format (step **h**). The subsequent fluorescent read-out gives a direct quantification of the amount of protein present in the sample as the SOMAmer forms a 1:1 complex with its target protein. Reprinted with permission from ASGCT, Mol Ther Nucleic Acids. 2014 Oct; 3(10): e201.

**Figure 9 ijms-21-04522-f009:**
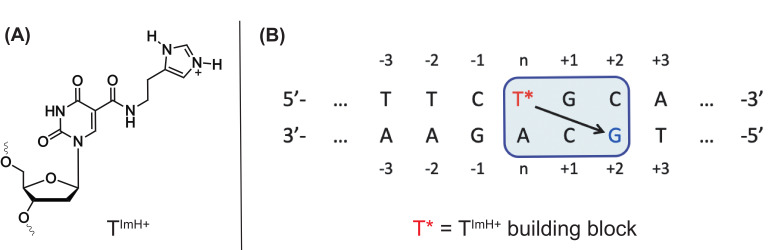
Structure of the T^ImH+^ building block (**A**) and example of the ‘stabilizing motif’ in a DNA duplex (**B**) with T* being the T^ImH+^ building block. The arrow indicates which guanine is involved via its Hoogsteen side in stabilizing hydrogen bonding with the imidazolium moiety.

**Figure 10 ijms-21-04522-f010:**
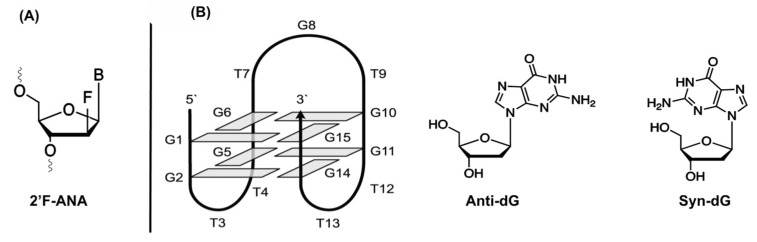
Structure of 2′F-ANA nucleosides (**A**); Structure of the thrombin binding aptamer with an edge-loop G-quadruplex conformation and the anti- and syn-oriented deoxyguanosine (**B**). Reprinted (adapted) with permission from *Chemistry Central Journal, 2014,* 8, 19—Published by Springer Nature. Creative Commons. License link available online: https://creativecommons.org/licenses/by/2.0. (accessed on 6 June 2020)

**Figure 11 ijms-21-04522-f011:**
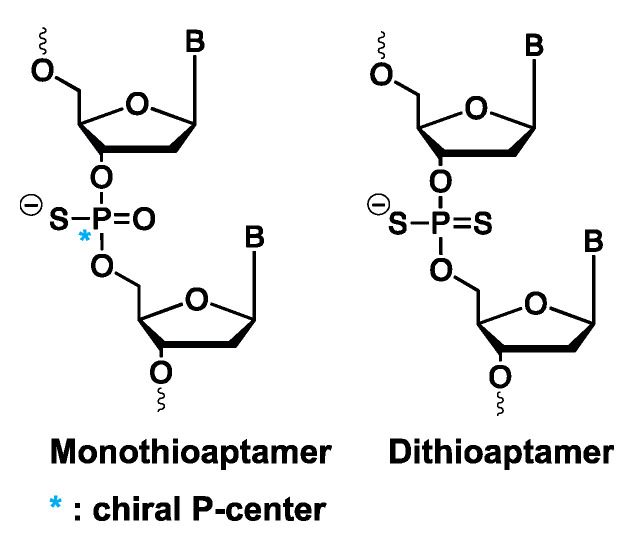
Structure of mono- and dithioaptamers.

**Figure 12 ijms-21-04522-f012:**
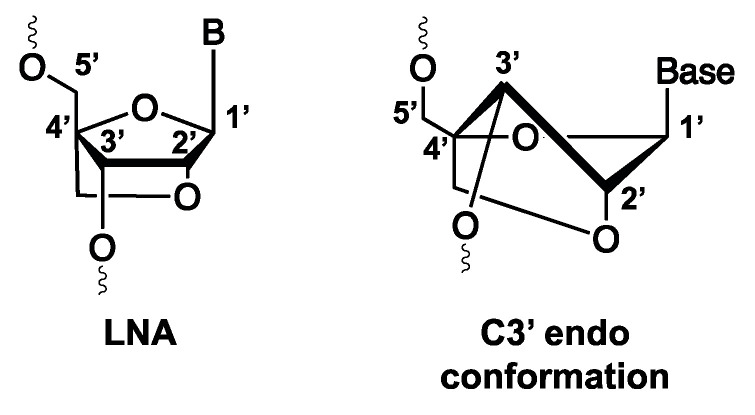
Structure of LNA (left) and the C3′ endo conformation (right).

**Figure 13 ijms-21-04522-f013:**
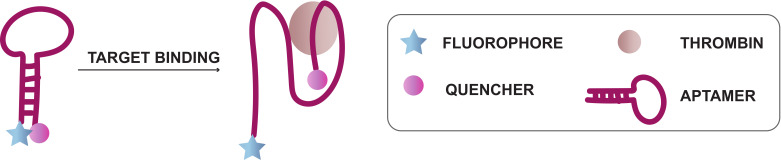
Principle of the anti-thrombin binding aptamer as molecular beacon.

**Figure 14 ijms-21-04522-f014:**
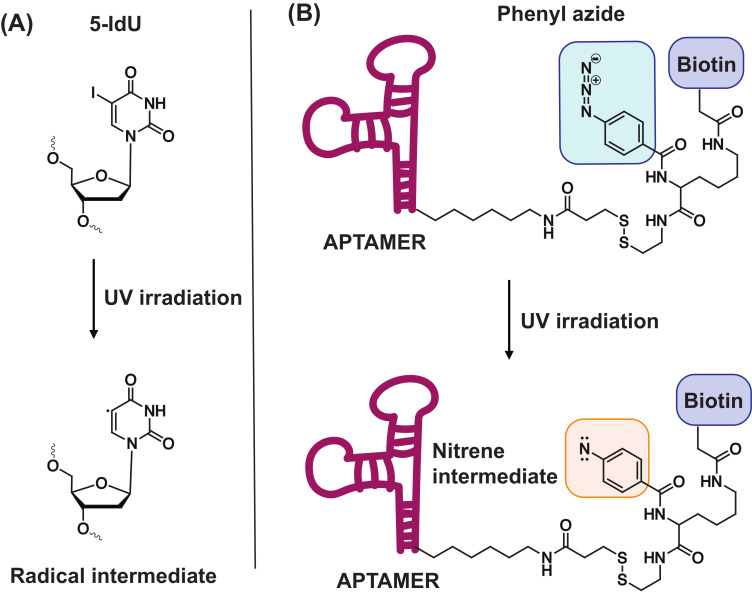
Structure of the crosslink reagents and reactive species formed upon UV irradiation of 5-iodo-2′-deoxyuridine (**A**) and phenyl azide (**B**).

**Figure 15 ijms-21-04522-f015:**
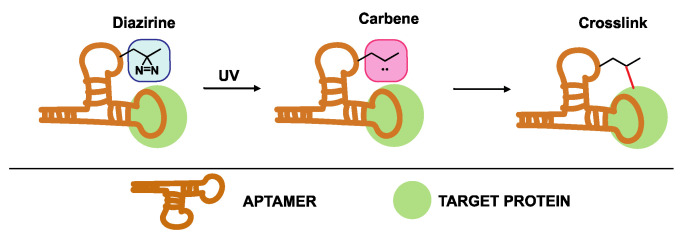
Principle of UV activated crosslinking between diazirine-modified aptamers and target proteins.

**Figure 16 ijms-21-04522-f016:**
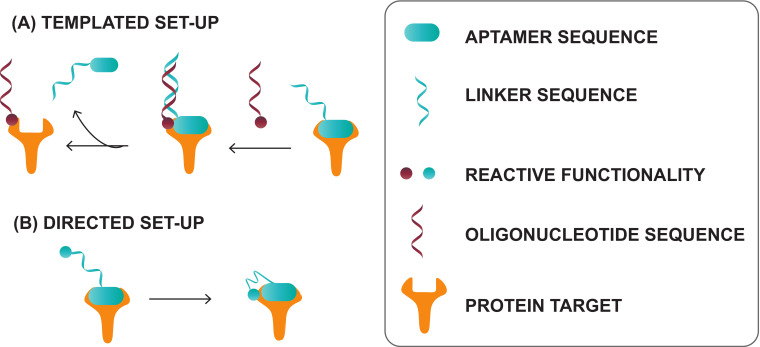
Principle of the templated directed ligation (**A**) and the directed non-templated ligation strategy (**B**). Reprinted (adapted) with permission from *Chem. Sci.,* 2016, 7, 2157–2161—Published by the Royal Society of Chemistry. Creative Commons. License link available online: https://creativecommons.org/licenses/by/3.0/legalcode (accessed on 8 June 2020).

**Figure 17 ijms-21-04522-f017:**
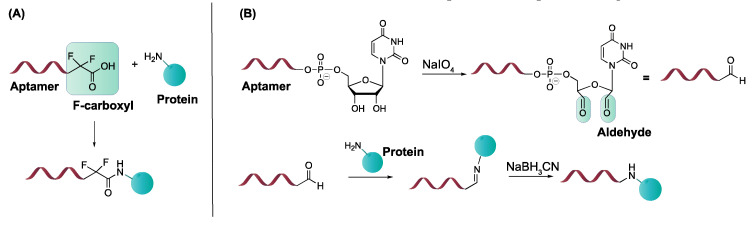
Templated directed ligation with an F-carboxyl-modified aptamer (**A**) and an aldehyde-modified aptamer (**B**).

**Table 1 ijms-21-04522-t001:** Overview of aptamer modifications discussed in this review. dZ—6-amino-5-nitro-3-(1′-β-D-2′-deoxyribofuranosyl)-2(1H)-pyridone; dP—2-amino-8-(1′-β-D-2′-deoxyribofuranosyl)-imidazo-[1,2-a]-1,3,5-triazin-4(8H)-one; Ds—7-(2-thienyl)-imidazo[4,5-b]pyridine; Px—2-nitro-4-propynyl-pyrrole; 2′ F-ANA—2′-Fluoro arabino nucleic acid; LNA—locked nucleic acid; 5-IdU—5-iodo deoxyuridine; B—nucleobase.

**Modifications For Non-Covalent Target Binding**
C5 modified nucleotides	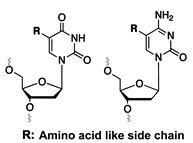	dZ-dP	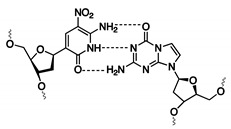
dDs-dPx	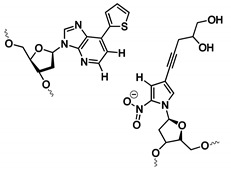	Thioaptamer	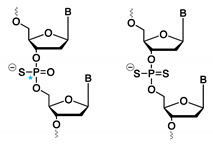
2′ F-ANA	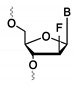	LNA	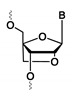
**Modifications For Covalent Target Trapping**
5′-IdU	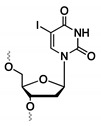	Diazirine	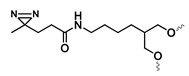
Aldehyde	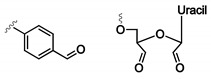	F-carboxyl	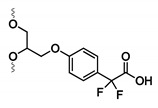
Phenyl azide	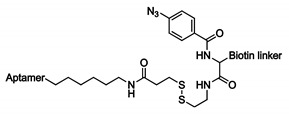

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
