# Peer review of "Chemical Modification of Aptamers for Increased Binding Affinity in Diagnostic Applications: Current Status and Future Prospects"

_ijms, 2020, doi:10.3390/ijms21124522_

Round 1
Reviewer 1 Report
Very clear, complete, interesting and rigourous perspective on aptamer chemical modification.
the paper is excellent and reviews in a very complete way chemical strategies to improve aptamers' performance in binding affinity and selectivity, whose scientific soundness has already been checked as they refer to published papers of high quality. The interest is clear, the authors explain very well aptamers advantages versus antibodies and clearly address immediate challenges to tackle for their use in diagnosis.
For completeness and if the authors consider them relevant in the introduction they might mention biosensing transducers other than electrochemical, such as gravimetric/nanoviscosity QCM-D and thermal resistance (HTM).
Author Response
First, we would like to thank the reviewer for the positive feedback on our manuscript. We further appreciate the suggestion to mention biosensing transducers other than electrochemical in the introduction. Despite the fact that modified aptamers are not yet used in this context, the latter applications demonstrate the versatility and potential of aptamers in a wide range of diagnostic applications. Therefore, we decided to mention the versatility of aptamer-based biosensors in the introduction (line 33-34). We have also included a reference to a recently published article (review, February 2020), which describes the wide range of aptamer-based biosensing strategies in more detail.
Reviewer 2 Report
The manuscript by Elskens et al. (ijms-844626) makes a great contribution to the aptamer knowledge by thoroughly reviewing how aptamers are modified for better diagnostic applications. The manuscript is very well organized into sections and subsections. The flow of the entire article is excellent. The vast amount of the reviewed information is accurate and current. I feel this article is very timely as it provides readers an insight into how aptamers and their modifications can potentially help mediate the global health challenge we are currently facing.
I do not have any recommended changes in the context.
However, I do want to point out that some figures resolution can be improved. I am not sure how much editing the IJMS office is going to perform on the graphs. In particular, the text in various figures is harder to see clearly, even on a high-resolution computer screen. I suggest the authors work with the IJMS editorial office on improving image qualities.
Thank you very much for this informative review article.
Author Response
We would like to thank the reviewer for the very positive and encouraging feedback on our manuscript and for pointing out that the resolution of some figures can be improved. We noticed that the quality of the figures in the submitted manuscript (as found on the MDPI website) does not correlate with the quality of the original file. Therefore, we contacted the Assistant Editor and the problem should be solved upon resubmission.
Reviewer 3 Report
The authors have provided an extremely well written and very current and comprehensive critical review of a variety of methodologies that may be employed to optimise aptamer reagents for potential diagnostic and therapeutic use. They are to be congratulated for a near faultless piece of writing which was a pleasure to read. I have just a few very minor suggested grammatical corrections / alternatives for the authors to consider :
LINE 69 : 'can allow to increase the' would read better as 'can allow an increase in binding....'
LINE 72 : 'consisting out of 20...' change to 'with an option of '
LINE 96 : 'are not yet used' change to 'have not yet been used'
LINES 125-6 : 'In what follows' change to ' In this review '
LINE 127 : 'on which we elaborate' change to 'relevant to this review'
LINE 189 : remove the word 'out'
LINE 190 : remove 'In general' .Start following sentence with The aptamer sequences normally encompass'
LINE 201 : 'are removed ; hereby' should read ' are removed, thereby'
LINE 203 : Insert the words 'for example' between 'has' and 'already' and delete 'e.g.' change 'resulted' to 'resulting'
LINE 215 : remove 'Next to colorimetric detection' and start sentence with 'Electrochemical' and insert 'also' between 'have' and 'been'.
LINE 308 : change 'In what follows' to 'Herein'
LINE 340 : change 'invigorates' to 'favours'
LINE 359 : Why brackets around 'diagnostic'?
LINE 393 : remove 'In this prospect' start sentence with 'Thus, further diversification..'
LINE 419 : require a comma after 'DNA molecule'
LINE 446 : change 'Next' to 'In addition'
LINE 447 : Remove 'that has been described' and insert documented' between 'One' and 'strategy'
LINE 518 : remove 'On the other hand' and replace with 'Conversely'
LINE 692 : 'act' should be plural 'acts'
LINE 705 : remove 'In what follows' replace with 'In this section'
LINE 706 : change word order thus 'our area of focus will be directed reactions as they...'
LINES 754-55 : I think the following might read better ' .. there is still scope for greater refinements and thus more successful future exploitation of their full potential as truly versatile reagents'
Author Response
First, we would like to thank the reviewer for the very positive and encouraging feedback on our manuscript and for suggesting grammatical corrections / alternatives. We corrected the manuscript accordingly as the corrections / alternatives improve the quality of the manuscript.
Reviewer 4 Report
In this manuscript, Elskens et al. presented an expansive literature compilation of the chemical modification strategies applied to aptamers. Because of the immense body of research, the authors chose to focus on the utility of such modifications in the development of diagnostic tools –as opposed to therapeutic tools, which has been recently reviewed in IJMS (Int. J. Mol. Sci. 2017, 18(8), 1683; https://doi.org/10.3390/ijms18081683).
The authors start their narrative with an introduction of aptamers and a comparative analysis between aptamers and antibodies, favoring the former for diagnostics. The authors went on and briefly presented the SELEX procedure and divided the chemical modifications into pre- and post-SELEX modifications. In the former the authors described nucleotides modifications, which they cautioned must be polymerase compatible –the authors explain that this is possible by modifying the C5 position in pyrimidine and N7 position in purine bases, ideally concurrent with the use of mutated DNA polymerases with high fidelity regarding modified nucleotides incorporation. In the post-SELEX procedures, the authors didn’t cite any example (page 4, last paragraph), but later caught up on. In section 3, the authors divided the chemical modifications into non-covalent and covalent target interactions. In the introductory paragraph of the former, they described 3 categories: truncation, multiple aptamer conjugation, and other specific modifications either in pre or post-SELEX setup (last paragraph page 5). The latter turned out including 5 specific modifications: (1) extended genetic alphabet (dP-dZ, and dDs-dPx), (2) amino acid like side chain modification (SOMAmers and imidazole-tethered thymine), (3) 2’-Fluoro-arabino-NA (2’-F-ANA), (4) mono and dithioaptamers, and 5 LNA-modified aptamers. In the modified aptamers for covalent target trapping, the authors explained that this strategy mostly involves UV-activated crosslinking and gave examples of specific modifications (5-IdU, 5’-end phenyl azide modification, and diazirine modification), but also introduced a novel approach which involves template-directed ligation between the aptamer and the target protein.
At the end of each modification’s paragraph and where applicable, the authors cited the latest advancement in diagnostic applications. The authors also nicely illustrated the described modifications with figures, although the quality of these figures was generally poor. Finally, the authors concluded with a concise recapitulative paragraph, in which they emphasized on the usefulness of the aptamer modification strategies for the development of innovative diagnostic tools in health and disease.
In general, the topic is interesting, the manuscript is well thought and timely. I only have a couple minor comments that, in my opinion, if taken into account, may increase the readability by the general audience.
- The manuscript may benefit from the addition of a flow-chart schematic in which the authors summarize the described chemical modifications.
- The authors may want to re-structure their sections. Indeed, going to the 4th subsection level is not recommended (3.2.3.1 and 3.2.3.2). As a suggestion, I would present solely “non-covalent target binding” as section 3, and the covalent binding in a new section 4. I would also split the pre-SELEX vs post-SELEX subsection into two subsections and give the post-SELEX paragraph its fair share –at least by enumerating the chemical modifications that fall into this category.
Other minor issues:
- Line 107: the authors wrote: “the DNA library should first be converted into RNA sequences via in vitro reverse transcription”. Please remove “reverse” as DNA to RNA copying is just a transcription.
- Ref 5 is wrongly attributed, its authors are different from those to whom the citation is attributed. Furthermore, it is irrelevant in the cited context. Please replace or remove.
- Line 130: The authors interestingly mentioned a combinatory approach in which both pre- and post-SELEX modifications are applied for increased selectivity and binding affinity. Could the authors cite specific examples, if any, from the literature? If not, could the authors comment on a plausible methodology?
- The authors may want to swap Figures 4 and 5 (and the text that describes them), because the introduced dZ-dP before dDs-dPx (line 308).
- SOMAmers was mentioned in line 372 before it was spelled out in line 381.
- Lines 411-427: The authors may want to include this nice description of SOMAmers in the legend of Figure 7.
- The authors may want to consider the following paper DOI: 1039/C5MB00045A(Paper) Mol. BioSyst., 2015, 11, 1260-1270, and potentially revise their statement about “no diagnostic tools incorporating LNA-modified aptamers have been developed yet” (line 586).
Misspelling and grammatical errors:
- Line 48: grammatical: “remediate this, these procedures…” can be replaced with “remediate this draw back, the procedures…”
- Line 88: an overview is provided of the available… I would move “is provided” to the end of the sentence.
- Line 93: “for the purpose,” should be for the purpose of…
- Line 123: unnecessary comma
- Line 126: Ref. 32 doesn’t belong to the sentence.
- Line 143: “structural requirements that modified nucleotides should obey…”
- Line 162: “In this case,” instead of “In that case,”
- Line 182: (often downsizing by truncation).
- Line 204: “a 56-mer” instead of “an 56-mer”
- Line 310: missing a closing parenthesis after “triazin-4(8H)-one”.
- Line 327: please specify to which protein the 100-fold increase in binding affinity belongs to.
- Line 391: “emphasize” instead of “emphasizes”
- Line 504: “2’-fluoro” the “2” is with the title not with the subsection numbering.
- Line 537: “warrant” instead of “warrants”
- Line 661: grammatical: “successfully” instead of “successful”
- Reference 132 is published in 2004 not 2005.
- Etc.
Author Response
Reviewer comment: In this manuscript, Elskens et al. presented an expansive literature compilation of the chemical modification strategies applied to aptamers. Because of the immense body of research, the authors chose to focus on the utility of such modifications in the development of diagnostic tools –as opposed to therapeutic tools, which has been recently reviewed in IJMS (Int. J. Mol. Sci. 2017, 18(8), 1683; https://doi.org/10.3390/ijms18081683).
Answer: First, we would like to thank the reviewer for his thorough, detailed and positive feedback on our manuscript. We decided to also include the reference indicated above by the reviewer concerning the use of modified aptamers in a therapeutic point of view, as it nicely illustrates the complementarity between both reviews.
Reviewer comment: At the end of each modification’s paragraph and where applicable, the authors cited the latest advancement in diagnostic applications. The authors also nicely illustrated the described modifications with figures, although the quality of these figures was generally poor.
Answer: We noticed that the quality of the figures in the submitted manuscript (as found on the MDPI website) does not correlate with the quality of the original file. Therefore, we contacted the Assistant Editor and the problem should be solved upon resubmission.
Reviewer comment:The manuscript may benefit from the addition of a flow-chart schematic in which the authors summarize the described chemical modifications.
Answer: We included a flow-chart (figure 3) visualizing the modifications that are discussed in this review, either incorporated via a pre- and/or post-SELEX strategy. It is however not evident to identify specific modifications that are solely incorporated via the pre-SELEX approach. Therefore, we divided the flow chart in two main categories, in which the green boxes are typical post-SELEX modifications, while the modifications in the purple boxes can be introduced both before (pre) or after (post) the SELEX process.
Reviewer comment:The authors may want to re-structure their sections. Indeed, going to the 4th subsection level is not recommended (3.2.3.1 and 3.2.3.2). As a suggestion, I would present solely “non-covalent target binding” as section 3, and the covalent binding in a new section 4. I would also split the pre-SELEX vs post-SELEX subsection into two subsections and give the post-SELEX paragraph its fair share –at least by enumerating the chemical modifications that fall into this category.
Answer: We thank the reviewer for these constructive suggestions. We have decided to indeed present “non-covalent target binding” as section 3 (from line 208 onwards), and the “covalent binding” in a new section 4 from line 676 onwards), as suggested by the reviewer. Moreover, we also followed the advice to enumerate the chemical modifications that fall into the category of post-SELEX modifications. However, we respectfully disagree to split the pre-SELEX vs post-SELEX section in two different subsections, since the majority of discussed modifications can be introduced both in a pre- and post-SELEX set-up. Therefore, a clear modification-based distinction between the two categories is difficult to be made and we believe the flow of the article will benefit from keeping the section as a whole. However, we do hope that the flow-chart we included (figure 3) is visually helping the reader to see the correlation between pre- and post-selex incorporation.
Other minor issues:
Line 107: the authors wrote: “the DNA library should first be converted into RNA sequences via in vitro reverse transcription”. Please remove “reverse” as DNA to RNA copying is just a transcription.
We removed the word reverse.
Ref 5 is wrongly attributed, its authors are different from those to whom the citation is attributed. Furthermore, it is irrelevant in the cited context. Please replace or remove.
Thank you for pointing this out. We replaced the reference with a reference to a review concerning the use of aptamer-based biosensors for antibiotic detection.
Line 130: The authors interestingly mentioned a combinatory approach in which both pre- and post-SELEX modifications are applied for increased selectivity and binding affinity. Could the authors cite specific examples, if any, from the literature? If not, could the authors comment on a plausible methodology?
We explained three specific examples that combine pre- and post-SELEX modifications in order to optimize aptamer properties (Line 142 – 152)
The authors may want to swap Figures 4 and 5 (and the text that describes them), because the introduced dZ-dP before dDs-dPx (line 308).
Instead of swapping the figures and the text that describes them, we introduced dDs-dPx prior to dZ-dP.
SOMAmers was mentioned in line 372 before it was spelled out in line 381.
At the first mention, the word SOMAmers was removed (line 451)
Lines 411-427: The authors may want to include this nice description of SOMAmers in the legend of Figure 7.
We added the description in the legend.
The authors may want to consider the following paper DOI: 1039/C5MB00045A(Paper) Mol. BioSyst., 2015, 11, 1260-1270, and potentially revise their statement about “no diagnostic tools incorporating LNA-modified aptamers have been developed yet” (line 586).
We would like to thank the reviewer for this suggestion. After evaluating this article, we changed our statement and referred to the article in question. Indeed, the immobilization of an LNA containing aptamer on a streptavidin-coated biosensor is described for the qualitative and selective analysis of target binding (his-tagged CD37) by bio-layer interferometry. Moreover, as this article represents another example of the successful combination of both a pre-SELEX (LNA-aptamer selection) and a POST-SELEX (truncation) methodology, this was also mentioned in the text.